# The IGF-Independent Role of IRS-2 in the Secretion of MMP-9 Enhances the Growth of Prostate Carcinoma Cell Line PC3

**DOI:** 10.3390/ijms242015065

**Published:** 2023-10-11

**Authors:** Haruka Furuta, Yina Sheng, Ayaka Takahashi, Raku Nagano, Naoyuki Kataoka, Claire Marie Perks, Rachel Barker, Fumihiko Hakuno, Shin-Ichiro Takahashi

**Affiliations:** 1Department of Animal Sciences and Applied Biological Chemistry, Graduate School of Agricultural and Life Sciences, The University of Tokyo, Tokyo 113-8654, Japan; harukafuruta0507@gmail.com (H.F.); shengyn526@gmail.com (Y.S.); ayk006100@gmail.com (A.T.); nagano-raku882@g.ecc.u-tokyo.ac.jp (R.N.); akataoka@g.ecc.u-tokyo.ac.jp (N.K.); 2IGFs & Metabolic Endocrinology Group, Learning & Research Building, Translational Health Sciences, Bristol Medical School, University of Bristol, Bristol BS8 1TH, UK; claire.m.perks@bristol.ac.uk (C.M.P.); mdrmh@bristol.ac.uk (R.B.)

**Keywords:** IRS-2, MMP-9, secretion, prostate cancer cells, PC3

## Abstract

Insulin receptor substrate-2 (IRS-2), a substrate of the insulin-like growth factor (IGF)-I receptor, is highly expressed in the prostate cancer cell line, PC3. We recently demonstrated that extracellular signal-regulated kinase (Erk1/2), a kinase downstream of IGF signaling, is activated in PC3 cells under serum starvation, and this activation can be inhibited by IRS-2 knockdown. Here, we observed that adding an IGF-I-neutralizing antibody to the culture medium inhibited the activation of Erk1/2. Suppression of Erk1/2 in IRS-2 knockdown cells was restored by the addition of a PC3 serum-free conditioned medium. In contrast, the IRS-2-silenced PC3 conditioned medium could not restore Erk1/2 activation, suggesting that IRS-2 promotes the secretion of proteins that activate the IGF signaling pathway. Furthermore, gelatin zymography analysis of the conditioned medium showed that matrix metalloproteinase-9 (MMP-9) was secreted extracellularly in an IRS-2 dependent manner when PC3 was cultured under serum starvation conditions. Moreover, MMP-9 knockdown suppressed Erk1/2 activation, DNA synthesis, and migratory activity. The IRS-2 levels were positively correlated with Gleason grade in human prostate cancer tissues. These data suggest that highly expressed IRS-2 activates IGF signaling by enabling the secretion of MMP-9, which is associated with hyperproliferation and malignancy of prostate cancer cell line, PC3.

## 1. Introduction

Insulin-like growth factors (IGFs) induce a variety of bioactivities, including cell growth and survival in many cell types [1]. These bioactivities have linked the IGF signaling system with cancer risk, tumorigenesis, tumor growth, and survival [2,3,4]. In general, the binding of IGF to specific IGF-I receptors (IGFIR) on the plasma membrane activates its intrinsic tyrosine kinase activity [5,6]. The activated IGFIR phosphorylates substrates, including insulin receptor substrates (IRSs) [7]. Phospho-tyrosine residues in IRSs are recognized by Src Homology 2 (SH2)-domain-containing signaling molecules such as Grb2 or the p85 PI 3-kinase regulatory subunit. In particular, the recruitment of Grb2 to IRSs is believed to activate the Ras/Raf/Mek cascade, resulting in extracellular signal-regulated kinase (Erk1/2) activation. Ras/Raf/Mek or PI 3-kinase cascade activation is required for IGF bioactivity, including cell growth [8,9].

IGFIR is commonly expressed in human cancers. Although its association with patient outcomes is debated upon, numerous reports have indicated that expression levels of IGFIR are associated with poor prognosis [4]. IRS-1/-2 is also expressed in some cancer cell lines, and its high expression has roles in cancer initiation and progression [10]. Overexpression of either IRS-1 or -2 has resulted in transformation of human MCF10A cells [10]. Similarly, the transgenic expression of IRS-1 or -2 in the mouse mammary gland causes mammary hyperplasia and tumorigenesis, accompanied by upregulation of the Akt and Erk1/2 pathways [10,11]. In addition, depletion of IRS-2 significantly diminished tumor growth and metastasis in various organs of mice [11,12], thus indicating that IRS-1/2 can serve as a positive regulator of cancer progression.

Matrix metalloproteinases (MMPs) mediate homeostasis of the extracellular environment by degrading the extracellular matrix and basement membrane [13,14]. MMPs foster invasion and spread by disrupting extracellular matrix barriers, but they also affect cellular signaling via several routes [15]. MMPs are also universal features of carcinoma progression associated with tumor angiogenesis, invasion, and metastasis [16]. The ability of MMPs to degrade the extracellular matrix contributes to the redevelopment of the surroundings and potential metastatic sites for cancer cells [17]. Thus, accumulating evidence shows a link between MMPs and cancer progression. However, the connection between IGF signaling and MMPs remains unknown.

Prostate cancer growth is highly dependent on IGF signaling, and over-activation of IGF signaling results in the transformation of prostate epithelial cells [18,19,20,21]. The expression levels of IRS-2 have been examined in human prostate tumors and several prostate cancer cell lines. IRS-2 expression levels are low in LNCap cells, which represents an indolent form of prostate cancer, while IRS-2 expression levels are high in PC3 and DU145 cells, which are highly aggressive prostate cancer cell types [12]. In particular, we have reported that in PC3 cells, IRS-2 protein levels are maintained by USP9X, which protects IRS-2 from degradation as a deubiquitinase [22]. These data support the idea that high expression of IRS-2 increases the activity of IGF signaling and thereby plays a role in establishing and maintaining prostate cancer. However, we have previously reported that IRS-2 was highly expressed and Erk1/2 was activated, even under serum-free conditions without IGF-I stimulation, in PC3 cells; this overactivation of Erk1/2 is required for anchorage-independent cell growth in PC3 [22]. Additionally, IRS-2 knockdown in PC3 cells significantly suppressed Erk1/2 activation, suggesting that IRS-2 increased Erk1/2 activation via an unknown mechanism. Thus, in this study, we investigated the mechanism involved in Erk1/2 activation under serum-free conditions by IRS-2 and found an IGF-independent role of IRS-2 in secreting MMP-9.

## 2. Results

### 2.1. Inhibition of IGF-I or IGFIR Decreased Erk1/2 Phosphorylation under Serum-Free Conditions in PC3 Cells

To investigate how Erk1/2 is activated by serum starvation, PC3 cells were serum-starved for 1, 3, 6, 10, or 24 h. Erk1/2 phosphorylation rapidly decreased within 1 h after serum depletion but was re-phosphorylated 3 h later (Figure 1A). However, Akt phosphorylation did not increase after serum depletion. To test the possibility that IGF-I in the medium may cause IGFIR/Erk1/2 pathway activation, PC3 cells were treated with a neutralizing antibody against IGF-I. This treatment decreased Erk1/2 phosphorylation under serum-free conditions (Figure 1B). In addition, small interfering RNA (siRNA) knockdown of IGF-I also decreased Erk1/2 phosphorylation under serum-free conditions (Figure 1C), suggesting that IGF-I, produced in PC3, plays an important role in Erk1/2 phosphorylation. Next, we examined whether IGFIR activation is required for Erk1/2 phosphorylation under serum-free conditions. We inhibited IGFIR activation via an IGFIR tyrosine kinase inhibitor, BMS754807, and assessed the Erk1/2 phosphorylation. Erk1/2 phosphorylation was blocked by adding the IGFIR inhibitor (Figure 1D). Moreover, knockdown of IGFIR decreased Erk1/2 phosphorylation under serum-free conditions (Figure 1E, Lane 13–18 vs. Lane 1–6). This suggests that the IGF-I produced by PC3 cells causes Erk1/2 phosphorylation through IGFIR activation. Erk1/2 phosphorylation was also impaired by IRS-2 knockdown (Figure 1E, Lane 7–12 vs. Lane 1–6).

### 2.2. Decrease of Erk1/2 Phosphorylation in IRS-2 Knockdown Cells Was Rescued by Culturing with Conditioned Medium from PC3 Cells

Next, we stimulated IRS-2 knockdown PC3 cells with a serum-free conditioned medium of PC3 cells. The experimental plan is shown in Figure 2A. Two days before the experiment, PC3 cells were transfected with either scramble siRNA or IRS-2 siRNA and were cultured in a serum-free medium for 24 h. The conditioned medium was collected from both transfected groups and were labeled as serum-free conditioned medium SFCM-ctrl or SFCM-IRS-2KD. Next, IRS-2 knockdown PC3 cells and normal PC3 cells were stimulated with SFCM-ctrl or SFCM-IRS-2KD for 1, 3, 6 h. When IRS-2 knockdown cells were cultured in serum-free medium (SFM), Erk1/2 phosphorylation was reduced, compared with the control cells (Figure 2B, Lane 4–6 vs. Lane 1–3). In contrast, when IRS-2 knockdown cells were cultured in SFCM-ctrl, the reduction in Erk1/2 phosphorylation was completely restored (Figure 2B, Lane 7–9 vs. Lane 4–6). However, when IRS-2 knockdown cells were cultured in SFCM-IRS-2KD, this restoration was impaired (Figure 2B, Lane 10–12 vs. Lane 7–9).

In addition, when IGFIR was knocked down, SFCM-ctrl did not induce Erk1/2 phosphorylation (Figure 2C, Lane 11–15), even though SFCM-ctrl did induce Erk1/2 phosphorylation when IRS-2 was knocked down (Figure 2C, Lane 6–10), indicating that IGFIR is vital for Erk1/2 activation.

### 2.3. MMP-9 Secretion from PC3 Cells Is Required for Erk1/2 Activation under Serum-Free Conditions

The above data suggest that something is secreted into the conditioned medium in an IRS-2 dependent manner, which activates the IGFIR/Erk1/2 pathway. We hypothesized that IGF-I could serve as a strong candidate for being secreted into medium. However, both IGF-I and IGF-II were undetectable in the serum-free conditioned medium (SFCM). qPCR analysis could detect the *Igf1* mRNA, but this level was increased (not decreased) by IRS-2 knockdown (KD) (Appendix A). These data suggest that another factor, other than IGF-I, is responsible for the activation of the IGFIR/Erk1/2 pathway. It is well known that some IGF-binding proteins (IGFBPs) regulate the binding of IGFs to IGFIR. It is possible that IGFBPs, which enhance the binding of IGFs to IGFIR, were secreted in an IRS-2 dependent manner. Thus, we examined IGFBP levels in the conditioned medium of PC3 cells. IGFBP2, IGFBP3, and IGFBP4 were detected in the SFCM (Appendix A). The protein levels of IGFBP2 and IGFBP3 increased in SFCM by IRS-2 KD and the protein level of IGFBP4 was decreased by IRS-2 KD. As the protein levels of candidate factors in SFCM are expected to be decreased by IRS-2KD, we excluded IGFBP2 and IGFBP3 as potential candidates. Additionally, the levels of IGFBP4 were decreased by IRS-2 KD, which is known to suppress the IGF-I activity—a completely opposite function expected by the potential candidate for this study; thus, it was excluded as well. Next, we tested the involvement of proteins that would alter the IGFBPs’ protein levels. Several proteases have been reported to cleave IGFBPs, allowing the release of IGFs from IGFBPs to activate IGFIR [20,23]. Thus, we tested the involvement of proteases in the conditioned medium through the pharmacological inhibition of proteases. The serine/cysteine protease inhibitors Pefabloc SC, EGTA, and EDTA suppressed Erk1/2 activation under serum-free conditions (Figure 3A,B), while pepstatin A (aspartic protease inhibitor) did not (Figure 3C). It has been reported that serine/cysteine protease inhibitors inactivate MMPs, and that MMPs cleave IGFBPs [24]. Thus, MMPs are possible candidate factors that regulate IGF-I activity. Next, we assessed the quantity of MMPs in the conditioned medium. A gelatin zymography assay is a method to detect the gelatinase activity of MMPs. The conditioned media from PC3 cells or IRS-2-silenced PC3 cells were subjected to SDS-PAGE using a gelatin-containing gel. Gelatinase activity was detected in the serum-free conditioned medium of PC3 cells, and this activity was suppressed in the IRS-2-silenced PC3 cells (Figure 3D). The molecular weight of approximately 90 kDa with gelatinase activity suggested that it might be MMP-9. On the other hand, a band of around 70 kDa, which corresponds to the molecular weight of MMP-2, could not be detected. Therefore, immunoblotting analysis of the conditioned medium using an anti-MMP-9 antibody revealed that the extracellular MMP-9 protein level was decreased by IRS-2 knockdown, even though the total protein levels in the conditioned medium were not changed by IRS-2 KD (Figure 3E). These data strongly indicated that IRS-2 is required for MMP-9 secretion. Next, we knocked down MMP-9, and examined Erk1/2 activation. Erk1/2 phosphorylation was blocked by MMP-9 knockdown (Figure 3F). If MMP-9 can regulate Erk1/2 phosphorylation by degrading IGFBPs, the knockdown of MMP-9 is expected to increase IGFBPs levels. However, MMP-9 knockdown did not increase IGFBP2, IGFBP3, or IGFBP4 levels (Appendix A).

### 2.4. IRS-2 Is Required for MMP-9 Secretion from the PC3 Cells

Next, we examined how IRS-2 increased extracellular MMP-9 protein levels under serum-free conditions. MMP-9 transcription levels were decreased when IRS-2 was knocked down in PC3 cells (Figure 4A). Immunostaining analysis using an MMP-9 antibody revealed that MMP-9 was stained in a punctate pattern in control PC3 cells (Figure 4B, upper). However, when IRS-2 was knocked down, MMP-9 accumulated in the cytosolic structure of the PC3 cells (Figure 4B, lower, white arrows). Green fluorescent intensity in the microscopic area was also significantly increased in IRS-2 KD PC3. This structure was partially colocalized with F-actin (Figure 4C). These data indicate that the decrease in extracellular MMP-9 protein levels in IRS-2 knockdown PC3 was not caused by the decrease in the amount of MMP9 mRNA but by the accumulation of MMP9 in cells due to the inhibition of secretion.

As IRS-2 is known to be a significant mediator of IGF-I signaling, we examined the possibility that the inhibition of MMP-9 secretion in IRS-2 knockdown cells was caused by suppression of IGF-I signaling. Treatment with BMS754807 (IGFIR inhibitor) or PD98059 (MEK inhibitor) did not show any decrease in MMP-9 levels but a slight increase in the extracellular levels of MMP-9 was observed (Figure 4D). Moreover, intracellular cytosolic accumulation of MMP-9 as observed in IRS-2 KD cells was not observed in the cells treated with neither inhibitor, as assessed using immunostaining analysis (Figure 4E). Thus, inhibition of IGF-I signal did not inhibit MMP9 secretion and inhibition of MMP-9 secretion in IRS-2 knockdown cells was not caused by suppression of IGF-I signaling.

### 2.5. IRS-2 Expression Levels Are Positively Correlated with the Gleason Grade in Prostate Cancer Tissues

Next, we examined that effects of MMP9 knockdown on cell proliferation and cell migration ability. MMP-9 or IRS-2 knockdown significantly suppressed thymidine incorporation into DNA under serum-free conditions (Figure 5A). MMP-9 or IRS-2 knockdown reduced migration, as shown by the scratch assay (Figure 5B).

We recently discovered that IRS-2 is not highly expressed in androgen-dependent prostate cancer LNCaP cells [22]. In these cells, we could neither detect MMP-9 secretion nor Erk1/2 activation (Appendix A). These data strongly suggest that IRS-2 may contribute to the malignant transformation of prostate cancer. Thus, the expression levels of IRS-2 and MMP-9 were examined in human prostate cancer tissues (Figure 5C). Notably, IRS-2 levels were positively correlated with the Gleason grade (Figure 5D). However, no correlation could be established between MMP-9 levels and the Gleason grade (Figure 5E).

The forced expression of IRS-2 in LNCaP did not enhance MMP-9 secretion or Erk1/2 activation (Appendix A), suggesting that IRS-2 expression is required, but is insufficient for Erk1/2 activation under serum-starvation conditions. The other prostate cancer DU145 cells, which are malignant similar to PC3, expressed high levels of IRS-2 similar to PC3 but did not secrete MMP-9 (Appendix A) [12,25]. In Du145 cells, it is possible that IRS-2 might secrete a protein different from MMP-9, resulting in increased malignancy.

## 3. Discussion

In the present study, we identified a novel role of IRS-2 in promoting the secretion of MMP-9, which causes activation of the IGF signaling pathway in human prostate cancer cells, PC3. The secretagogue activity of IRS-2 is independent of the IGF signal activation, since it is not inhibited by IGF signal inhibitors. It is presumed that IGF-I, which is produced by PC3 cells, is highly dense in the vicinity of the cell, without diffusion into the culture supernatant because IGF-I is attached to the cell membrane. These data indicate that the IRS-2-dependent secretion of MMP-9 activated IGF-I, which existed in an inactive state in the vicinity of the cell, thereby transducing signals through IGFIR kinase to phosphorylate Erk1/2. It might be a unique system for malignant cancer cells that highly activates IGF signals with only a small amount of IGF-I produced by itself. IRS-2 might contribute to tumor malignancy by promoting the secretion of factors. Factors secreted by IRS-2 might differ from cell to cell, since the IRS-2-dependent secretion of MMP-9 was not observed in the malignant prostate cancer cells DU145. We are currently trying to isolate factors that are secreted in DU145 in an IRS-2-dependent manner. This is possibly why the Gleason grade was correlated with the IRS-2 expression level but not with the MMP-9 expression level.

As described above, PC3 and DU145 are highly aggressive and malignant prostate cancer cell lines. In contrast, LNCap cells represent the most indolent form of prostate cancer. IRS-2 protein levels are high in PC3 and DU145 (Appendix A) [12], but lower in LNCap compared to PC3 cells (Appendix A) [12], indicating that IRS-2 protein levels are correlated with the malignancy of prostate cancer cell lines. In addition, a positive correlation was observed between IRS-2 protein levels and Gleason grade in human prostate cancer tissue (Figure 5C). However, in TCGA database analysis, IRS-2 gene mutations or gene amplification were observed in only a few percent of prostate cancer tissues [26,27,28], suggesting no correlation of IRS-2 gene expression with stages of prostate cancer. Actually, we previously reported that the IRS-2 protein interacts with deubiquitinase, USP9X, and this interaction plays important roles in the maintenance of IRS-2 protein levels and that IRS-2 mRNA levels are higher in LNCap than in PC3 [22]. Therefore, IRS-2 protein levels are not determined by just gene expression but also by protein degradation. The data that no association was observed in TCGA analysis (gene level), but a correlation was observed in immunohistochemistry (protein level), are therefore quite novel insights, which are hard to validate in the online databases.

IGF signaling is initiated by the interaction of IGFIR with free IGFs; however, most IGFs are bound to IGFBPs in the serum, with only a small amount found in the free form [23,24,25,29]. IGFBPs regulate the half-life of IGFs and interaction of IGF-I with IGF-I receptors. In addition, IGF activity is regulated by certain proteases that cleave IGFBPs [30,31,32,33,34]. MMPs are one such protease that are inactivated by serine/cysteine protease inhibitors [35], and they degrade IGFBPs, thus regulating IGF signaling [36,37]. Consistent with these previous reports, we identified MMP-9 to be involved in Erk1/2 pathway regulation. Moreover, MMP-9 is one of the most complex forms of MMPs with multiple pathophysiological functions, the most common of which is its ability to degrade the ECM [38]. Degradation of the ECM is likely to release IGFBP-3 originally bound to ECM [37], which may explain the slight decrease in IGFBPs in the conditioned medium observed in this study. Although we failed to clarify the exact regulatory mechanism of MMP-9 in the IGFIR pathway, MMP-9 may possibly affect other IGFBPs.

Extracellular MMP-9 activity and expression was downregulated by IRS-2 knockdown, suggesting that the secretion of MMP-9 vesicles was blocked by IRS-2 knockdown, which is the first evidence that IRS-2 can regulate protein secretion. Previous studies have shown that IRS-2 interacts with various proteins, but not through recognition of tyrosine phosphorylation [39,40,41], and have demonstrated the novel roles of several IRS-2-associated proteins [22,42,43,44]. In our study, we observed that the actin polymerization state changed in IRS-2 knockdown cells, blocking the movement of vesicles to exocytotic sites and serving as a barrier until instant depolymerization [45]. This suggests that IRS-2 interacts with proteins to regulate the actin polymerization state.

This study has a few limitations. First, the IRS-2-forced expression in LNCaP did not enhance MMP-9 secretion. Thus, IRS-2 expression was not sufficient for MMP-9 secretion, suggesting the existence of additional unknown secretary hormones with important roles in Erk1/2 activation. Further analysis is required to elucidate the molecular mechanisms. Second, the Erk1/2 phosphorylation level stimulated with SFCM-IRS-2KD was greater than that stimulated with SFM, which can be attributed to two possibilities. In these experiments, the SFCM was collected from the medium in which the cells were cultured for 24 h in advance. Therefore, the MMP-9 protein amount in the SFCM-ctrl was higher than that of the SFM-ctrl at 6 h. In addition, because the IRS-2 knockdown was incomplete, the MMP-9 protein amount in the SFCM-IRS2KD was not zero. Thus, the Erk1/2 phosphorylation was greater than the IRS-2 KD cells with SFM. Therefore, in future experiments, the time of collection of the medium should be maintained the same and knockout experiments should be performed to further validate the results.

In conclusion, this study demonstrated for the first time that IRS-2 could modulate protein secretion, which markedly expands our understanding of the role of IRS-2 in IGF signal pathway activation and cancer cell proliferation. It is assumed that this regulatory mechanism of protein secretion would lead to a more thorough understanding of the protein interactions of IRS-2 and IRS-2-associated proteins. In addition, the finding that MMP-9 plays a key role in IGF signal pathway activation in PC3 cells could provide a novel strategy for improved prostate cancer diagnosis and treatment. However, future studies are needed to further strengthen our findings.

## 4. Materials and Methods

### 4.1. Materials

Dulbecco’s modified Eagle’s medium (DMEM) and phosphate-buffered saline (PBS) were purchased from Nissui Pharmaceutical CO. (Tokyo, Japan). Fetal bovine serum (FBS) was obtained from Sigma Aldrich (St. Louis, MO, USA). Penicillin and streptomycin were obtained from Banyu Pharmaceutical CO. (Ibaraki, Japan). Anti-IRS-2 (sc-390761) and anti-IGFIRb (sc-713) antibodies were purchased from Santa Cruz Biotechnology (Dallas, TX, USA). Anti-Akt (9272S), anti-pAkt (Ser473; 9271S), anti-Erk1/2 (9102S), and anti-pErk1/2 (9101S) antibodies were purchased from Cell Signaling Technology (Danvers, MA, USA). Anti-MMP-9 antibody (ab38898) was purchased from Abcam (Cambridge, UK). Pepstatin A was purchased from Sigma-Aldrich (St. Louis, MO, USA) and Pefabloc SC was obtained from Cayman Chemical (Ann Arbor, MI, USA). Gelatin was purchased from Sigma-Aldrich. PD98059 was purchased from New England Biolabs (Ipswich, MA, USA), and BMS754807 was purchased from Selleck chemicals (Houston, TX, USA). IGF-I neutralizing antibodies (Sm1.2) were purchased from Merck Millipore (Middlesex Country, MA, USA). TRITC (tetramethylrhodamine isothiocyanate)-conjugated phalloidin was obtained from Sigma. Hoechst33342 was obtained from Molecular Probes (Eugene, OR, USA). Horseradish peroxidase (HRP)-conjugated secondary anti-rabbit and anti-mouse IgG antibodies were obtained from GE Healthcare (Pittsburgh, PA, USA). Enhanced chemiluminescence (ECL) reagents were obtained from PerkinElmer Life Science (Boston, MA, USA).

### 4.2. Cell Culture

The prostate cancer PC3 cells were kindly provided by Dr. Akio Matsubara (Hiroshima University). LNCaP was kindly provided by Dr. Eijiro Nakamura (Kyoto University, Kyoto, Japan) [46]. The DU145 cells were purchased from the American Type Tissue Culture Collection. All cells were maintained at 37 °C in a humidified 5% CO_2_-controlled atmosphere in DMEM supplemented with 10% FBS, 0.1% NaHCO_3_, 50 IU/mL penicillin, and 50 µg/mL streptomycin. Cells were serum-starved in DMEM supplemented with 0.1% bovine serum albumin (BSA) for 1–24 h.

### 4.3. Immunoblotting Analysis

Cell lysates were prepared with lysis buffer (50 mM Tris-HCl, pH 7.4, 1 mM EDTA, 1 mM EGTA, 150 mM NaCl, 50 mM NaF, 1% Triton X-100, 100 kallikrein-inactivating units/mL aprotinin, 20 mg/mL phenylmethanesulfonyl fluoride, 10 mg/mL leupeptin, 5 mg/mL pepstatin) and subjected to immunoblotting as described previously [43].

### 4.4. siRNAs

The siRNAs used in this study were obtained from Nippon Gene Material Corp. (Tokyo, Japan). The siRNAs comprised the following sequences: IGF-I #1, 5′-GACAGGGGCUUUUAUUUCA-3′; IGF-I #2, 5′-AGGUGAAGAUGCACACCAU-3′; IRS2 #1, 5′-UCGGCUUCGUGAAGCUCAA-3′; IRS2 #2, 5′-GGCUGAGCCUCAUGGAGCA-3′; IGFIR 5′-GGAGUUCAAUUGUCACCAU-3′; MMP-9 #1, 5′-GGCCAAUCCUACUCCGCCU-3′; MMP-9 #2, 5′-GGAGCCAGUUUGCCGGAUA-3′.

### 4.5. Transfection with Plasmids or siRNA

The expression plasmids were transfected into PC3 cells using polyethylenimine (PEI) as described previously [43]. The siRNAs were transfected into PC3 cells using Lipofectamine RNAiMAX (Life Technologies, Tokyo, Japan) via the reverse transfection method, following the manufacturer’s protocol.

### 4.6. Preparation of Conditioned Medium and Stimulation with Conditioned Medium

Preparation of the conditioned medium and stimulation with the serum-free conditioned medium (SFCM) is illustrated in Figure 2A. Two days before the experiments, cells were transfected with scrambled siRNA and human IRS-2. One day later, the medium was replaced with the serum-free medium. Twenty-four hours later, the medium was collected, centrifuged (4 °C, 1000× *g*, 5 min,) to remove cell debris, and concentrated 20 times using centrifugal filters (pore size: 3 kDa, Merck Millipore Ltd. Tullagreen, Carrigtwohill, Co Cork, Ireland) as serum-free conditioned medium (SFCM-ctrl, SFCM-IRS-2 KD). The PC3 cells were transfected with scrambled siRNA against the control, human IRS-2, and human IGFIR. Twenty-four hours later, cells were cultured in the fresh serum-free medium (SFM), SFCM-ctrl, or SFCM-IRS-2 KD for 1, 3, and 6 h. Cell lysates were prepared from each cell and subjected to immunoblotting analysis using indicated antibodies.

### 4.7. Gelatin Zymography Assay

The gelatin zymography was performed as follows: SDS polyacrylamide gels were copolymerized with gelatin (0.1%). Electrophoresis was performed using a mini gel slab apparatus with a constant voltage of 150 V at 4 °C until the dye reached the bottom of the gel. Following electrophoresis, the gels were washed in a wash buffer (50 mM Tris-HCl pH 7.5, 2.5% Triton X-100, 5 mM CaCl_2_, 1mM ZnCl_2_) for 30 min to remove SDS. Then, the zymograms were incubated overnight at 37 °C in an incubation buffer (50 mM Tris-HCl pH 7.5, 1% Triton X-100, 5 mM CaCl_2_, 1mM ZnCl_2_). The gels were then stained with Coomassie brilliant blue and destained with a buffer (49% methanol, 1% acetic acid). Areas of enzymatic activity appeared as clear bands on the blue background.

### 4.8. DNA Synthesis Assay

Cells were serum-starved for 24 h. DNA synthesis was measured using [^3^H] thymidine as described previously [47].

### 4.9. Wound Assay

PC3 cells transfected with siRNA were seeded into 12-well plates at 2.0 × 10^5^ cells/well and incubated for 24 h. Then, a sterile 200-μL plastic pipette tip was used to create a scratch wound across the PC3 monolayer, following which, the detached cells were removed by washing with cold phosphate-buffered saline (PBS; 137 mM NaCl, 2.68 mM KCl, 8.10 mM Na_2_HPO_4_ and 1.47 mM KH_2_PO_4_ pH 7.4) three times. The cells were then cultured in serum-free DMEM for 24 h and photographed at 0 and 24 h after scraping.

### 4.10. Real-Time PCR

Total RNA was extracted from cells using TRIzol Reagent (Invitrogen, Carlsbad, CA, USA), and cDNA synthesis was performed using ReverTra Ace qPCR RT Master Mix (Toyobo, Osaka, Japan). cDNA was subjected to qPCR using THUNDERBIRD Next SYBR qPCR Mix (Toyobo, Osaka, Japan) and Applied Biosystems StepOne Real-Time PCR System (Thermo Fisher Scientific, Waltham, MA, USA). Detailed protocols were followed according to the manufacturer’s instructions. Human β-actin (actb) mRNA was used as an internal control in all samples for data normalization. The primers comprised the following sequences: MMP-9 forward, 5′-TACTCGACCTGTACCAGCGAG-3′; MMP-9 reverse, 5′-GGAATGATCTAAGCCCAGCGC-3′; IGF1 forward, 5′-GGGGCTTTTATTTCACAAGCCC-3’; IGF1 reverse, 5′-CTTCTGGGTCTTGGGCATGT-3′; actb forward, 5′-TTCCTTCCTGGGCATGGAG-3′; and human actb reverse, 5′-GCAGTGATCTCCTTCTGCATC-3′.

### 4.11. Immunostaining Analysis

Primary antibodies (anti-MMP-9) and Alexa Fluor conjugated secondary antibodies (Life Technologies, Carlsbad, CA, USA) were diluted using a blocking buffer (3% BSA/PBS (−)). The nucleus was stained with Hoechst 33,342 and actin stress fiber was stained with TRITC-conjugated phalloidin (Sigma) by adding it to the secondary antibody. The cells were seeded on a cover slip. After removal of the medium, 4% paraformaldehyde/PBS (−) was added and the cells were immobilized at 25 °C for 15–20 min. After three washes with PBS (−), 0.25% Triton X-100/PBS (−) was added, and the treatment was carried out at 25 °C for 5 min. After three washes with PBS (−), a blocking buffer was added and incubated at 25 °C for 1 h. The primary antibody diluted in blocking buffer was added and incubated overnight at 4 °C. After washing with PBS (−), the secondary antibody and fluorescent dye diluted in blocking buffer was added and incubated at RT for 1 h. After washing three times with PBS (−), the cover glass was mounted on a slide glass using Vectashield (Vector Laboratories, Newark, CA, USA). The immunostaining images were captured using FV-3000 (Olympus, Tokyo, Japan). The fluorescence intensity was measured using ImageJ 1.53k.

### 4.12. Prostate Tissue Study

Clinical prostate samples were collected for the Prostate: Evidence for Exercise and Nutrition Trial (PrEvENT) at Southmead Hospital, Bristol, with full ethical approval (Ethics 14/SW/0056) [48]. Male patients provided their consent for prostate tissue samples, which was collected at the time of their surgeries, to be obtained. All samples were stored following the Human Tissue Act 2004 and the study was performed in accordance with the Declaration of Helsinki. The sections examined were surpluses of the diagnostic requirements. Patient clinical and pathological data including age at surgery, pre-surgery PSA level, Gleason and histological grade, and lymph node invasion were extracted and analyzed anonymously from medical records for this study.

The prostate tissues (n = 100) were formalin-fixed and paraffin-embedded, and 4 μm thick sections were cut using a microtome (Leica, Wetzlar, Germany) and collected on Tomo adhesive microscope slides (Matsunami, Bellingham, WA, USA). Immunohistochemistry was performed using a Ventana BenchMark ULTRA™ machine (Roche, Oro Valley, AZ, USA) according to the manufacturer’s protocol. Briefly, the sections were deparaffinized, pretreated with a cell conditioning step, and incubated with the primary antibody against IRS-2 (Thermo Fisher, Waltham, MA, USA; 1:5000). The sections were then counterstained with hematoxylin, dehydrated, and mounted in Clearium mounting medium (Leica). Slides were scored by two pathologists using a modified version of the Allred system, combining the proportion of tissue stained (on a scale of 1–5) with the staining intensity (on a scale of 1–3) to give a score out of 8 [49].

### 4.13. Statistical Analysis

Data are expressed as the mean ± standard error of the mean (S.E.M.). Comparisons between the two groups were performed using the Student’s *t*-test. Comparisons between more than two groups were performed using one-way analysis of variance (ANOVA). If the *p*-value obtained from the ANOVA was less than 0.05, the post-hoc tests indicated in each figure legend were performed. A value of *p* < 0.05 was considered statistically significant. All statistical calculations were performed using JMP^®^ Pro version 17 software (SAS Institute Inc., Cary, NC, USA).

For the prostate tissue study, normality was tested using Kolmogorov–Smirnov tests and then the protein abundance in relation to Gleason grade was analyzed using ANOVA with LSD post-hoc tests.

## Figures and Tables

**Figure 1 ijms-24-15065-f001:**
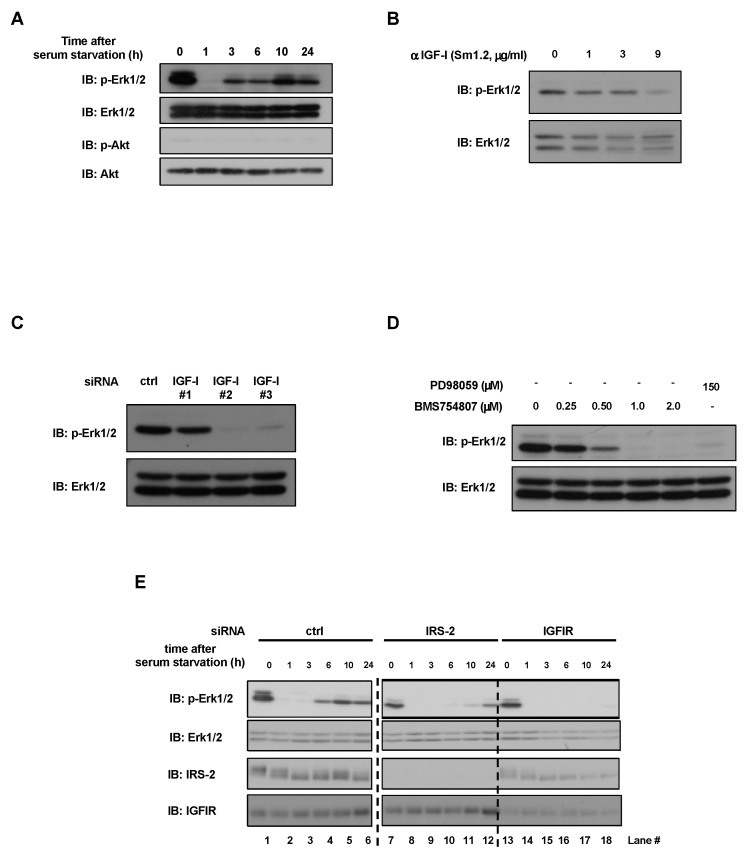
Effects of IGF-I or IGFIR inhibition on Erk1/2 phosphorylation under serum-free conditions in PC3 cells. (**A**): PC3 cells were serum-starved for the indicated time-period. Cell lysates were subjected to immunoblotting using the indicated antibodies. (**B**): PC3 cells were serum-starved for 24 h in the presence of an IGF-I-neutralizing antibody. Erk1/2 activation was examined by immunoblotting. (**C**): PC3 cells were transfected with short interfering RNA (siRNA) against scramble (ctrl) or IGF-I (#1, #2, #3). Each cell was serum-starved for 24 h, and Erk1/2 activation was examined. (**D**): PC3 cells were serum-starved in the presence of BMS754807 (IGFIR inhibitor) or PD98059 (MEK inhibitor). Erk1/2 activation was assessed by immunoblotting. (**E**): PC3 cells were transfected with siRNA against scramble (ctrl), IRS-2, or IGFIR. Each cell was serum-starved for the indicated time-period and cell lysates were prepared for immunoblotting analysis with the indicated antibodies. Lane numbers are shown below the blots. The experiments were performed multiple times independently, and a representative result is presented.

**Figure 2 ijms-24-15065-f002:**
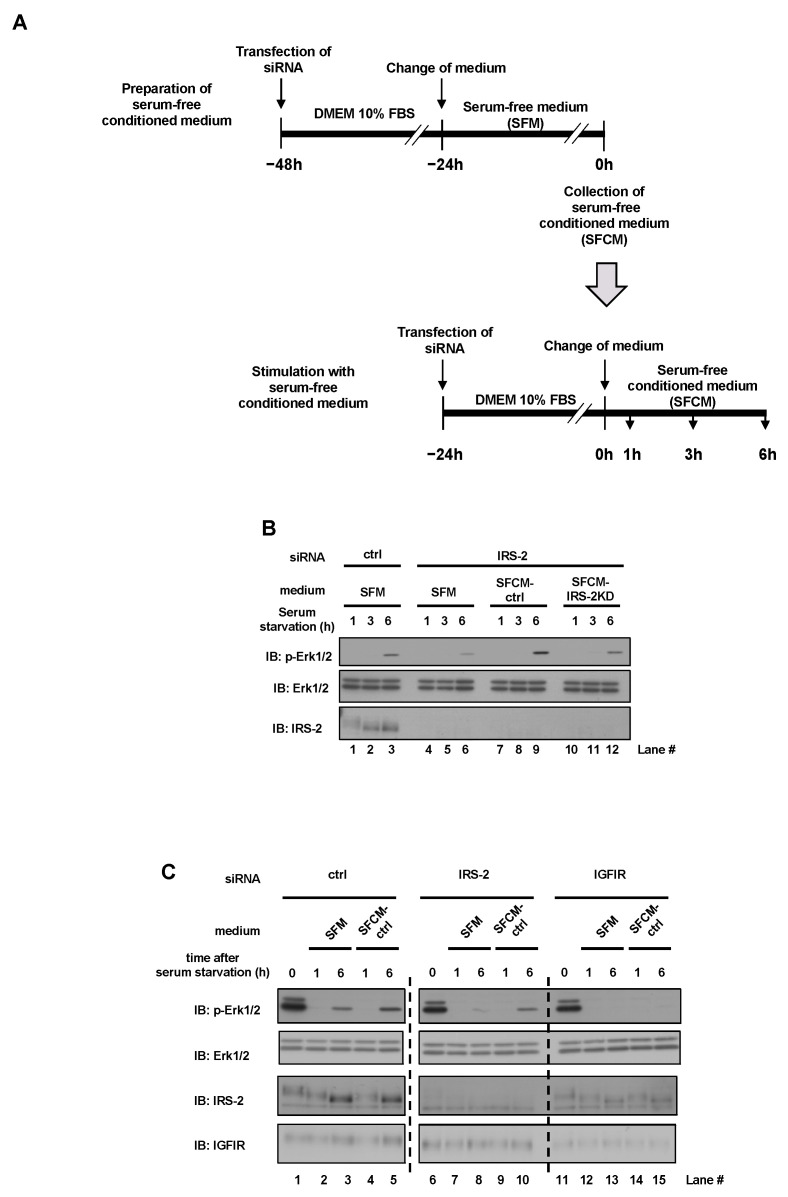
Effects of conditioned medium on Erk1/2 phosphorylation stimulation. (**A**): Experimental time course is illustrated. PC3 cells were transfected with scramble siRNA or IRS-2 siRNA and cultured in DMEM 10% FBS medium for 24 h. Then the cells were cultured in serum-free medium for an additional 24 h and the conditioned media (SFCM-ctrl or SFCM-IRS-2 KD) were collected. And then, IRS-2 knockdown PC3 cells or normal PC3 cells were cultured in SFCM-ctrl or SFCM-IRS-2 KD for indicated time periods. (**B**): PC3 cells were transfected with scramble siRNA (ctrl) or IRS-2 siRNA and were cultured in fresh serum-free medium (SFM), SFCM-ctrl, or SFCM-IRS-2 KD for 1, 3, or 6 h, and Erk1/2 activation was examined. Lane numbers are shown below the blots. (**C**): PC3 cells were transfected with scramble siRNA (ctrl), IRS-2 siRNA, or IGFIR siRNA and were cultured in SFM or SFCM-ctrl medium, and Erk1/2 activation was examined. Lane numbers are shown below the blots. The experiments were performed multiple times independently, and a representative result is presented.

**Figure 3 ijms-24-15065-f003:**
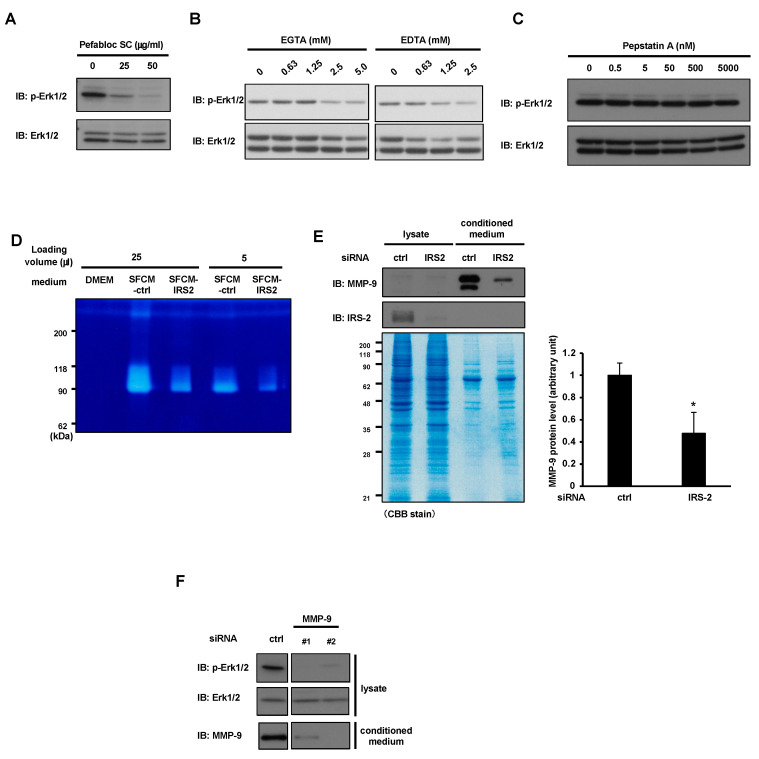
Identification of secretary proteins which activate the IGF signal system. PC3 cells were serum-starved for 24 h in the presence of the indicated concentrations of Pefabloc SC (**A**), EGTA, EDTA (**B**), or pepstatin A (**C**). Erk1/2 activation was examined by immunoblotting. (**D**): PC3 cells were cultured in a serum-free medium for 24 h. The conditioned medium (SFCM-ctrl) was collected. PC3 cells in which IRS-2 was knocked down were cultured in a serum-free medium for 24 h. The conditioned medium (SFCM-IRS-2KD) was collected. Serum-free DMEM, SFCM-ctrl, or SFCM-IRS-2KD was subjected to SDS-PAGE using gelatin-containing gel, and gelatinase activity was measured (gelatin zymography assay). (**E**): PC3 cells were transfected with scramble (ctrl) siRNA or IRS-2 siRNA, and cell lysates or conditioned medium in serum-free medium were collected. Lysates or conditioned medium were subjected to SDS-PAGE and immunoblotted with the indicated antibodies or underwent CBB staining. The amount of MMP-9 protein in the conditioned medium calculated from the immunoblotting data is shown on the right. Bar graphs are presented as fold-change of columns on the left (ctrl). Bar: mean ± S.E.M., * *p* < 0.05, n = 3, Student’s *t*-test. (**F**): PC3 cells were transfected with scramble (ctrl) siRNA or MMP-9 siRNA (#1, #2). Each cell line was cultured in a serum-free medium for 24 h, and the cell lysates were analyzed by immunoblotting with the indicated antibodies. The experiments were performed multiple times independently, and a representative result is presented.

**Figure 4 ijms-24-15065-f004:**
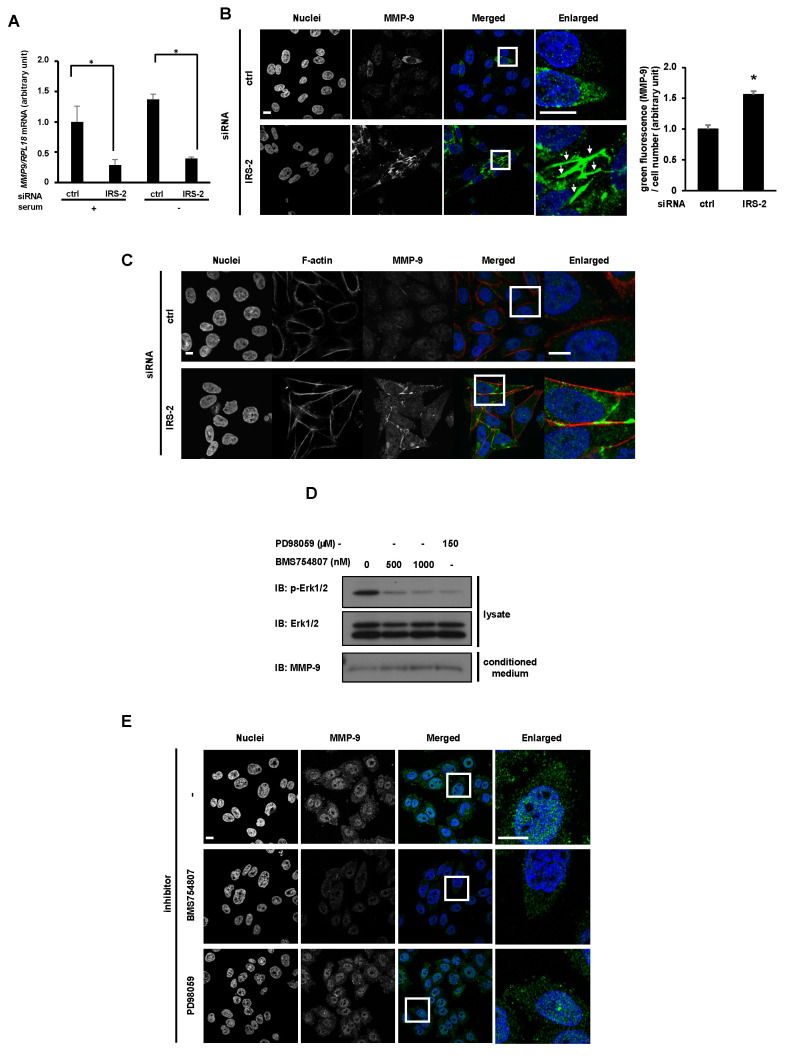
Analysis of IRS-2-mediated MMP-9 secretion. (**A**): PC3 cells were transfected with scramble (ctrl) siRNA or IRS-2 siRNA. The MMP-9 mRNA level was then examined by real-time PCR analysis. Bar graphs are presented as fold-change of columns on the far left (serum +, ctrl). Bar: mean ± S.E.M., * *p* < 0.05, n = 3, Student’s *t*-test. (**B**): PC3 cells were transfected with scramble (ctrl) siRNA or IRS-2 siRNA. Cells were cultured on the coverslip in a serum-free medium for 24 h. Cells were stained with anti-MMP-9 antibody (green) or Hoechst33342 (blue). Enlarged images of the white boxes were shown on the right. White arrows indicated MMP-9 accumulated structures. Scale bar: 10 µm. MMP-9 green fluorescence signal is shown on the right side of the graph. Bar graphs are presented as fold-change of columns on the left (ctrl). Bar: mean ± S.E.M., * *p* < 0.05, n = 15, Tukey–Kramer test. (**C**): PC3 or IRS-2 KD PC3 cells were serum-starved for 24 h and stained with anti-MMP-9 antibody (green), phalloidin (red) or Hoechst 33342 (blue). Enlarged images of the white boxes were shown on the right. Scale bar: 10 µm. (**D**): PC3 cells were serum-starved for 24 h in the presence of PD98059 (MEK inhibitor) or BMS754807 (IGFIR inhibitor), and Erk1/2 activation was examined. (**E**): PC3 cells were serum-starved for 24 h in the presence of the indicated inhibitors. Cells were stained with anti-MMP-9 antibody (green) or Hoechst 33342 (blue). Enlarged images of the white boxes were shown on the right. Scale bar: 10 µm. The experiments were performed multiple times independently, and a representative result is presented.

**Figure 5 ijms-24-15065-f005:**
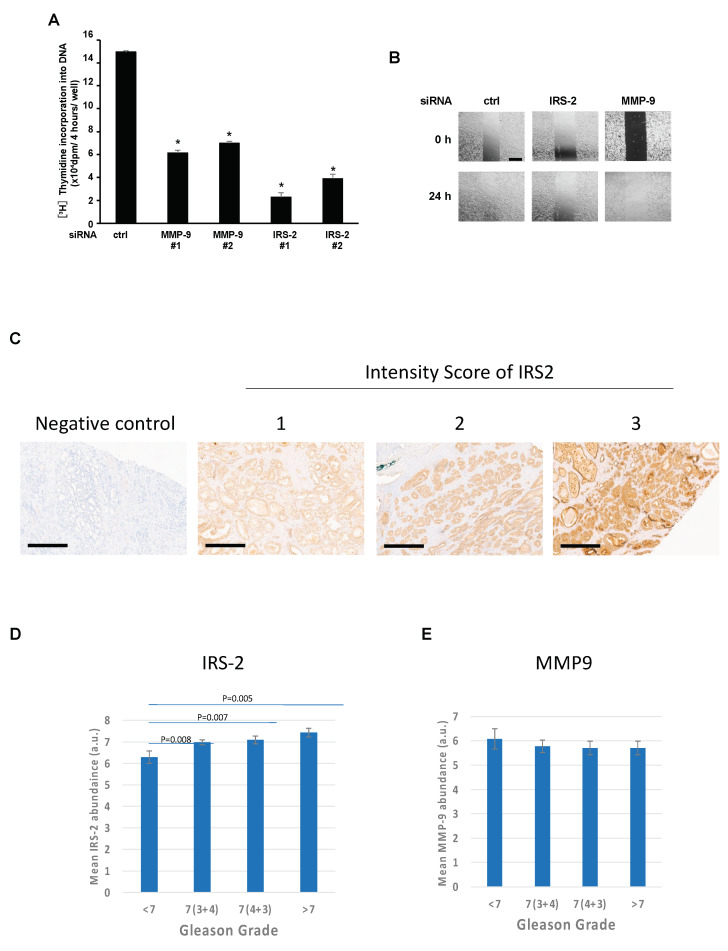
Relationship between IRS-2 expression and malignancy of prostate cancer. (**A**): PC3 cells were transfected with scramble (ctrl), MMP-9, or IRS-2 siRNA. The cells were subjected to DNA synthesis analysis. Bar graphs are presented as fold-change of columns on the left (ctrl). Bar: mean ± S.E.M., * *p* < 0.05, n = 3, Tukey–Kramer test. (**B**): PC3 cells were transfected with scramble (ctrl), MMP-9, or IRS-2 siRNA. Cells were subjected to a wound assay and migration activity was examined. Scale Bar; 1 mm and are applicable to all images in each panel. The experiments were performed multiple times independently, and a representative result is presented. (**C**): The expression of IRS-2 in human prostate cancer tissue was assessed by immunohistochemistry. Representative images of weak (1), moderate (2), and strong (3) expression. Scale Bar; 300 µm. (**D**,**E**): IRS-2 expression was significantly increased with the increasing Gleason grade, whereas MMP-9 expression was not significantly correlated with Gleason grade. Detailed analytical methods are described in Section 4.12. Bar: mean ± S.E.M., LSD post-hoc tests.

## Data Availability

All the original data for this manuscript are available upon reasonable request.

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
