# Peer review of "The IGF-Independent Role of IRS-2 in the Secretion of MMP-9 Enhances the Growth of Prostate Carcinoma Cell Line PC3"

_ijms, 2023, doi:10.3390/ijms242015065_

Round 1

Reviewer 1 Report

In this MS, authors suggested that highly expressed IRS-2 activates IGF signaling by the secretion of MMP-9, enhances proliferation and malignancy of PC3 cells. Despite interesting data, it has some concerns as follows:

1.     Show expression level of IRS-2 in LNCap, PC3 and DU145 cells. Why do you choose PC3 cells in your study?

2.     Show TCGA analysis on the role of IRS2 in several stages of prostate cancer tissues

3.     How about effect on MMP2 by IRS2 or ERK inhibitor or depletion

4.     How about effect of IRS-2 or ERK inibitor on BCL2 in PC3 cells after androgen deprivation therapy?

In this MS, authors suggested that highly expressed IRS-2 activates IGF signaling by the secretion of MMP-9, enhances proliferation and malignancy of PC3 cells. Despite interesting data, it has some concerns as follows:

1.     Show expression level of IRS-2 in LNCap, PC3 and DU145 cells. Why do you choose PC3 cells in your study?

2.     Show TCGA analysis on the role of IRS2 in several stages of prostate cancer tissues

3.     How about effect on MMP2 by IRS2 or ERK inhibitor or depletion

4.     How about effect of IRS-2 or ERK inibitor on BCL2 in PC3 cells after androgen deprivation therapy?

Reviewer 2 Report

The study titled "The IGF-Independent Role of IRS-2 in the Secretion of MMP-9 Enhances the Growth of Prostate Carcinoma Cell Line PC3" represents a significant contribution to our understanding of the role of IRS-2 in promoting MMP-9 secretion and its implications for the activation of the IGF signaling pathway in human prostate cancer cells, specifically PC3. The research is commendable not only for its well-structured and articulate presentation but also for the appropriateness of the methodology employed.

Minor revisions

Statistical Analysis- Please clarify how sample normality was assessed.

Line 263- Correct “independnt”  

The quality of the english presented is acceptable. 

Round 2

Reviewer 1 Report

much improved

ok

Reviewer 2 Report

My questions related to statistical analysis were not properly addressed. The normality of the data cannot be assessed through 'The experiments were performed multiple times independently, and a representative result is presented.' The statistical section in the materials and methods should be improved. If necessary, the authors should consult a specialist in this area.
